# Antimicrobial resistance, Extended-Spectrum β-Lactamase production and virulence genes in *Salmonella enterica* and *Escherichia coli* isolates from estuarine environment

**Saharuetai Jeamsripong** *, Mullika Kuldee, Varangkana Thaotumpitak, **Rungtip Chuanchuen**

Research Unit in Microbial Food Safety and Antimicrobial Resistance, Department of Veterinary Public Health, Faculty of Veterinary Science, Chulalongkorn University, Bangkok, Thailand

* saharuetai.j@gmail.com

**Data Availability Statement:** The raw data was available at https://figshare.com/articles/dataset/Antimicrobial_Resistance_Extended-Spectrum_-Lactamase_Production_and_Virulence_Genes_in_

## Abstract

The impact of antimicrobial resistance (AMR) on global public health has been widely documented. AMR in the environment poses a serious threat to both human and animal health but is frequently overlooked. This study aimed to characterize the association between phenotype and genotype of AMR, virulence genes and Extended-Spectrum β-Lactamase (ESBL) production from estuarine environment. The *Salmonella* ($n = 126$) and *E. coli* ($n = 409$) were isolated from oysters and estuarine water in Thailand. The isolates of *Salmonella* (96.9%) and *E. coli* (91.4%) showed resistance to at least one antimicrobial agent. Multidrug resistance (MDR) was 40.1% of *Salmonella* and 23.0% of *E. coli*. Resistance to sulfamethoxazole was most common in *Salmonella* (95.2%) and *E. coli* (77.8%). The common resistance genes found in *Salmonella* were *sul3* (14.3%), followed by *bla*$_{TEM}$ (11.9%), and *cmlA* (11.9%), while most *E. coli* were *bla*$_{TEM}$ (31.5%) and *tetA* (25.4%). The ESBL production was detected in *Salmonella* (1.6%, $n = 2$) of which one isolate was positive to *bla*$_{TEM-1}$. Eight *E. coli* isolates (2.0%) were ESBL producers, of which three isolates carried *bla*$_{CTX-M-55}$ and one isolate was *bla*$_{TEM-1}$. Predominant virulence genes identified in *Salmonella* were *invA* (77.0%), *stn* (77.0%), and *fimA* (69.0%), while those in *E. coli* isolates were *stx1* (17.8%), *lt* (11.7%), and *stx2* (1.2%). Logistic regression models showed the statistical association between resistance phenotype, virulence genes and ESBL production ($p < 0.05$). The findings highlighted that estuarine environment were potential hotspots of resistance. One Health should be implemented to prevent AMR bacteria spreading.

## Introduction

Antimicrobial resistance (AMR) has been recognized as one of the greatest challenges endangering the health of people, animals, and the environment. One Health approach has been applied for managing and controlling AMR at national and international levels. The Unites States Center for Disease Control and Prevention (U.S. CDC) estimated that greater than 2.8

Salmonella_enterica_and_Escherichia_coli_
Isolates_from_Estuarine_Environment/21761042

**Funding:** This study was supported by the Thailand
Science Research and Innovation (TSRI)
(CU_FRB640001_01_31_9). The funders had no
role in study design, data collection and analysis,
decision to publish, or preparation of the
manuscript.

**Competing interests:** The authors have declared
that no competing interests exist.

million people and at least 35,000 deaths are affected by AMR in the U.S. annually [1]. Without
a global response to AMR, it has been predicted that AMR could cause 10 million deaths annu-
ally by 2050 [2]. The global action plan to tackle AMR lists strengthening knowledge through
AMR surveillance as one of the important measures to address the global AMR issue [3]. How-
ever, AMR monitoring and surveillance in the environment is rather limited and not harmo-
nized due to several pathways responsible for AMR releasing to the environment. Therefore,
knowing the magnitude of the AMR in the environment is needed for estimating impact on
human and animal health.

*Salmonella* spp. is one of the most frequently isolated foodborne pathogens and a major
public health threat worldwide. Humans usually get infected with *Salmonella* by consumption
of contaminated food and water. Different food commodities, including poultry, swine, fish,
shellfish, and produce were linked to salmonellosis in humans [4]. Salmonellosis causes 93.8
human cases and almost 155,000 deaths annually [5]. *Salmonella* contains many virulence fac-
tors that play a crucial role in the ability to infect the host cells and propagate. *Salmonella* viru-
lence factors enhancing pathogenesis include *Salmonella* pathogenicity islands (SPIs),
*Salmonella* virulence plasmids, pili, and enterotoxin [6, 7]. Despite a particular public health
concern, knowledge of abundance of genotypic diversity of resistant and virulent *Salmonella*
isolated from the environment is still limited.

Bivalve mollusks serve as useful and practical bioindicators of environmental fecal contami-
nation. They have the capacity to accumulate nutrients, chemicals, and various microorgan-
isms [8]. *E. coli* has been used to monitor fecal contamination and AMR in bacteria from
food-producing animals for public purposes [9, 10]. Resistant *E. coli* can spread and transfer
their resistance determinants to inter- and intra-bacterial species. A previous metagenomic
analysis in untreated sewage revealed that multidrug resistance (MDR) bacteria were com-
monly found in waste disposed to the environment [11]. Therefore, the estuarine environment
is the area of particular concern due to the high diversity and abundance of resistant bacteria
that potentially pose a significant public health threat [12].

Extended-Spectrum β-Lactamases (ESBLs) are a group of enzymes that confer resistance to
penicillins, third and fourth generation cephalosporins, and monobactams [13]. The spread of
various types of ESBL-producing *E. coli* and *Salmonella* has been reported in different sectors,
including human, livestock animals, and aquaculture. The genes encoding of ESBL are associ-
ated with mobile genetic elements, which can horizontally transfer to other bacterial species
[14]. Infection of ESBL-producing bacterial pathogens in patients has been increasingly
reported to be associated with treatment failure and increase morbidity and mortality rates
due to limited effective antibiotics. Therefore, the objectives of this study were to examine phe-
notypic and genotypic AMR, virulence genes, and ESBL production, and to build statistical
models of the association between the most common resistance phenotype and other resis-
tance phenotype and genotypes, virulence genes, and ESBL production among *Salmonella* and
*E. coli* isolated from oyster and estuarine water samples.

## Materials and methods

### Bacterial strains

*Salmonella* (*n* = 126) and *E. coli* (*n* = 409) isolates were collected from stored collection strains
in the Department of Veterinary Public Health, Faculty of Veterinary Science, Chulalongkorn
University. All isolates were stored in 20% glycerol stock solution at -80˚C. The *Salmonella* iso-
lates were collected from oysters (*n* = 123) and estuarine water (*n* = 3), whereas the *E. coli* iso-
lates were retrieved from oysters (*n* = 250) and estuarine water (*n* = 159) samples. Oysters and
estuarine waters were collected monthly between April 2016 and March 2017 from Phang Nga

province in southern Thailand as previously described [15]. The oysters were wild caught, and not exposed to antimicrobials.

## Antimicrobial susceptibility testing

The agar dilution method was performed to determine minimum inhibitory concentrations (MICs) according to the Clinical and Laboratory Standard Institute [16]. Eight antimicrobials and their breakpoints were ampicillin (32 μg/ml), chloramphenicol (32 μg/ml), ciprofloxacin (4 μg/ml), gentamicin (8 μg/ml), streptomycin (32 μg/ml), sulfamethoxazole (512 μg/ml), tetracycline (16 μg/ml) and trimethoprim (16 μg/ml). *E. coli* ATCC 25922, *Pseudomonas aeruginosa* ATCC 27853 and *Staphylococcus aureus* ATCC 29213 were used for quality control. The multidrug resistance (MDR) was classified as resistance to at least three groups of antimicrobials.

## Detection of AMR gene

All isolates were tested for the presence of AMR genes including genes represented to ampicillin (*bla*$_{TEM}$), chloramphenicol (*catA*, *catB* and *cmlA*), quinolone (*qnrA*, *qnrB*, and *qnrS*), aminoglycosides (*acc(3)IV* and *aadA1*), streptomycin (*strA* and *strB*), tetracycline (*tetA* and *tetB*), sulfamethoxazole (*sul1*, *sul2*, and *sul3*), and trimethoprim (*dfrA1* and *dfrA12*) (Table 1). Conventional PCR was performed to detect most AMR genes, except genes corresponding to quinolone and sulfamethoxazole, which were used multiplex PCR. DNA templates of all *E. coli* and *Salmonella* were prepared using whole cell boiling technique [17]. Toptaq PCR Master Mix Kit (Merck, Munich, Germany) were followed as manufacturer's instruction. The PCR products were separated by gel electrophoresis using 1.5% agarose gel in 1X Tris-acetate/EDTA. Gels were stained with Redsafe™ Nucleic Acid Staining Solution (iNtRon Biotechnology, Seongnam, South Korea) and visualized PCR products under UV light using Omega Fluor™ gel documentation system.

## Phenotypic and genotypic detection of ESBL production

Disk diffusion method was used to examine ESBL production followed by CLSI standard [16]. The detection of ESBL production consists of screening and confirmation tests. Ceftazidime (30 μg), cefotaxime (30 μg), and cefpodoxime (10 μg) were used for initial screening. All isolates that showed resistance to at least one of cephalosporins were further confirmed using a combination disk diffusion method using cephalosporins combination with clavulanic acid. The positive ESBL production was interpreted by determining the difference of inhibition zone between solely cephalosporin and cephalosporin combine with clavulanic acid. The positive ESBL-production isolates were identified β-lactamases genes (*bla*$_{TEM}$, *bla*$_{SHV}$, *bla*$_{CMY-2}$, and *bla*$_{CTX-M-55}$). The *bla*$_{TEM}$ gene was examined using conventional PCR, while *bla*$_{SHV}$, *bla*$_{CMY-2}$, and *bla*$_{CTX-M-55}$ were using multiplex PCR with the specific primers as described in Table 2.

## Nucleotide sequence

PCR amplicons of positive ESBL production isolates were purified using GeneJET PCR purification kit (Thermo Fisher Scientific, Vilnius, Lithuania) and submitted for DNA sequencing (Bionics Co., ltd., Gyeonggi-Do, Republic of Korea). The result of the DNA sequence was blasted and aligned with references embedded in GenBank database available from the National Centre for Biotechnology Information (NCBI) (http://www.ncbi.nlm.nih.gov/BLAST) (accession number OQ282894-OQ282896).

**Table 1. Primer used and PCR condition for antimicrobial resistance genes.**

| Gene | Primer | Primer sequence | Denaturation | Cycle | Temperature and time (30 cycles) | | | Final | Amplicon size (bp) | Reference |
|------|--------|-----------------|--------------|-------|----------|-----------|-----------|-----------|------------|-----------|
| | | | | | Denaturation | Annealing | Extension | Extension | | |
| *catA* | catA-F | CCAGACCGTTCAGCTGGATA | 5 min at 94˚C | 30 | 45 s at 95˚C | 45 s at 55˚C | 10 s at 72˚C | 10 min at 72˚C | 454 | [18] |
| | catA-R | CATCAGCACCTTGTCGCCT | | | | | | | | |
| *catB* | catB-F | CGGATTCAGCCTGACCACC | 5 min at 94˚C | 30 | 45 s at 95˚C | 45 s at 55˚C | 10 s at 72˚C | 10 min at 72˚C | 461 | [18] |
| | catB-R | ATACGCGGTCACCTTCCTG | | | | | | | | |
| *cmlA* | cmlA-F | TGGACCGCTATCGGACCG | 5 min at 94˚C | 30 | 45 s at 94˚C | 45 s at 57˚C | 1 min at 72˚C | 5 min at 72˚C | 641 | [18] |
| | cmlA-R | CGCAAGACACTTGGGCTGC | | | | | | | | |
| *qnrA* | qnrA-F | AGAGGATTTCTCACGCCAGG | 10 min at 95˚C | 35 | 60 s at 95˚C | 60 s at 54˚C | 60 s at 72˚C | 10 min at 72˚C | 580 | [19] |
| | qnrA-R | TGCCAGGCACAGATCTTGAC | | | | | | | | |
| *qnrB* | qnrB-F | GGMATHGAAATTCGCCACTG | 10 min at 95˚C | 35 | 60 s at 95˚C | 60 s at 54˚C | 60 s at 72˚C | 10 min at 72˚C | 264 | [19] |
| | qnrB-R | TTTGCYGYYCGCCAGTCGAAC | | | | | | | | |
| *qnrS* | qnrS-F | GCAAGTTCATTGAACAGGGT | 10 min at 95˚C | 35 | 60 s at 95˚C | 60 s at 54˚C | 60 s at 72˚C | 10 min at 72˚C | 428 | [19] |
| | qnrS-R | TCTAAACCGTCGAGTTCGGCG | | | | | | | | |
| *aac(3)IV* | aac(3)IV-F | GTGTGCTGCTGGTCCACAGC | 3 min at 95˚C | 35 | 30 s at 94˚C | 30 s at 55˚C | 60 s at 72˚C | 10 min at 72˚C | 627 | [20] |
| | aac(3)IV-R | AGTTGACCCAGGGCTGTCGC | | | | | | | | |
| *aadA1* | aadA1-F | CTCCGCAGTGGATGGCGG | 5 min at 95˚C | 30 | 45 s at 95˚C | 45 s at 55˚C | 45 s at 72˚C | 10 min at 72˚C | 631 | [18] |
| | aadA1-R | GATCTGCGCGCGAGGCCA | | | | | | | | |
| *strA* | strA-F | TGGCAGGAGGAACAGGAGG | 15 min at 95˚C | 35 | 30 s at 94˚C | 1.5 min at 57˚C | 1.5 min at 72˚C | 10 min at 72˚C | 405 | [18] |
| | strA-R | AGGTCGATCAGACCCGTGC | | | | | | | | |
| *strB* | strB-F | GGCAGCATCAGCCTTATAATTT | 15 min at 95˚C | 35 | 30 s at 94˚C | 1.5 min at 57˚C | 1.5 min at 72˚C | 10 min at 72˚C | 470 | [21] |
| | strB-R | GTGGATCCGTCATTCATTGTT | | | | | | | | |
| *tetA* | tet(A)-F | GGCGGTCTTCTTCATCATGC | 5 min at 95˚C | 30 | 45 s at 94˚C | 1 min at 63˚C | 1 min at 72˚C | 10 min at 72˚C | 502 | [22] |
| | tet(A)-R | CGGCAGGCAGAGCAAGTAGA | | | | | | | | |
| *tetB* | tet(B)-F | CGCCCAGTGCTGTTGTTGTC | 5 min at 95˚C | 30 | 45 s at 95˚C | 45 s at 55˚C | 45 s at 72˚C | 10 min at 72˚C | 615 | [18] |
| | tet(B)-R | CGCGTTGAGAAGCTGAGGTG | | | | | | | | |
| *sul1* | sul1-F | CGGCGTGGGCTACCTGAACG | 10 min at 95˚C | 30 | 60 s at 95˚C | 60 s at 66˚C | 60 s at 72˚C | 10 min at 72˚C | 433 | [22] |
| | sul1-R | GCCGATCGCGTGAAGTTCCG | | | | | | | | |
| *sul2* | sul2-F | CGGCATCGTCAACATAACCT | 10 min at 95˚C | 30 | 60 s at 95˚C | 60 s at 66˚C | 60 s at 72˚C | 10 min at 72˚C | 721 | [22] |
| | sul2-R | TGTGCGGATGAAGTCAGCTC | | | | | | | | |
| *sul3* | sul3-F | TGTGCGGATGAAGTCAGCTC | 10 min at 95˚C | 30 | 60 s at 95˚C | 60 s at 66˚C | 60 s at 72˚C | 10 min at 72˚C | 244 | [22] |
| | sul3-R | GCTGCACCAATTCGCTGAACG | | | | | | | | |
| *dfrA1* | dfrA1-F | GGAGTGCCAAAGGTGAACAGC | 8 min at 94˚C | 32 | 60 s at 95˚C | 70 s at 55˚C | 10 min at 72˚C | 10 min at 72˚C | 367 | [23] |
| | dfrA1-R | GAGGCGAAGTCTTGGGTAAAAAC | | | | | | | | |
| *dfrA12* | dfrA12-F | TTCGCAGACTCACTGAGGG | 8 min at 94˚C | 32 | 60 s at 95˚C | 70 s at 55˚C | 10 min at 72˚C | 10 min at 72˚C | 330 | [18] |
| | dfrA12-R | CGGTTGAGACAAGCTCGAAT | | | | | | | | |

## Detection of virulence genes

Virulence genes of *Salmonella*, including invasin (*invA*), fimbrial protein (*fimA*), and entero-toxin (*stn*) genes were observed (Table 3). Heat-labile toxin (*lt*), heat-stable toxin (*st*), STEC (*stx1* and *stx2*) and EPEC for attaching and effacing protein (*eae*) were examined in all *E. coli* isolates. Most of virulence genes were detected using conventional PCR. The detection of *stx1* and *stx2* genes was performed by multiplex PCR.

## Statistical analyses

Descriptive statistics were performed to identify prevalence of resistance phenotype and geno-type, resistance pattern, MDR, virulence genes, and ESBL production of *E. coli* and *Salmonella*

**Table 2. Primer used and PCR condition for extended-Spectrum β-Lactamase genes.**

| Gene | Primer | Primer sequence | Denaturation | Cycle | Temperature and time | | | Final Extension | Amplicon size (bp) | Reference |
|---|---|---|---|---|---|---|---|---|---|---|
| | | | | | Denaturation | Annealing | Extension | | | |
| $bla_{TEM}$ | blaTEM-F | TTAACTGGCGAACTACTTAC | 3 min at 94˚C | 25 | 60 s at 94˚C | 60 s at 50˚C | 60 s at 72˚C | 10 min at 72˚C | 247 | [22] |
| | blaTEM-R | GTCTATTTCGTTCATCCATA | | | | | | | | |
| $bla_{SHV}$ | blaSHV-F | AGGATTGACTGCCTTTTTG | 3 min at 94˚C | 25 | 60 s at 94˚C | 60 s at 50˚C | 60 s at 72˚C | 10 min at 72˚C | 393 | [22] |
| | blaSHV-R | ATTTGCTGATTTCGCTCG | | | | | | | | |
| $bla_{CMY-2}$ | blaCMY-2-F | GACAGCCTCTTTCTCCACA | 3 min at 94˚C | 25 | 60 s at 94˚C | 60 s at 60˚C | 60 s at 72˚C | 10 min at 72˚C | 1000 | [22] |
| | blaCMY-2-R | GGACACGAAGGCTACGTA | | | | | | | | |
| $bla_{CTX-M}$ | blaCTX-M-F | CGATGTGCAGTACCTAA | 3 min at 94˚C | 25 | 60 s at 94˚C | 60 s at 60˚C | 60 s at 72˚C | 10 min at 72˚C | 585 | [24] |
| | blaCTX-M-R | AGTGACCAGAATCAGCGG | | | | | | | | |
| $bla_{CTX-M}$ group 1 | blaCTX-M group1-F | TTAGGAARTGTGCCGCTGYA | 10 min at 94˚C | 30 | 40 s at 94˚C | 40 s at 60˚C | 60 s at 72˚C | 7 min at 72˚C | 688 | [25] |
| | blaCTX-M group1-R | CGATATCGTTGGTGGTRCCAT | | | | | | | | |

isolates. Logistic regression analysis was used to examine the association among AMR, virulence genes and ESBL production. The dependent variable was the highest resistance rate, and independent variables included resistance genes, resistance phenotype, virulence genes, ESBL production and MDR. A $p$-value and confidence intervals of regression analyses were adjusted for potential correlated data within type of sample (oysters and estuarine waters) using robust variant estimator. Univariate analysis was performed to screen for potential significance of predictors. Forward selection and backward elimination were used to select potential candidates for multivariable analysis. Final regression models of *E. coli* and *Salmonella* were received based on $p < 0.05$ and likelihood ratio test. All statistical analyses were performed using Stata 14.0 (StataCorp, TX, USA). Two-sided hypothesis tests were used with 5% of significant level.

**Table 3. Primer used and PCR condition for virulence gene in *E. coli* and *Salmonella* isolates.**

| Gene | Primer | Primer sequence | Denaturation | Cycle | Temperature and time | | | Final Extension | Amplicon size (bp) | Reference |
|---|---|---|---|---|---|---|---|---|---|---|
| | | | | | Denaturation | Annealing | Extension | | | |
| *lt* | lt-F | TCTCTATGCATACGGAG | 5 min at 95˚C | 30 | 60 s at 95˚C | 60 s at 55˚C | 60 s at 72˚C | 10 min at 72˚C | 322 | [26] |
| | lt-R | CCATACTGATTGCCGCAATT | | | | | | | | |
| *st* | st-F | TGCTAAACCAGTAGAGTCTTCAAAA | 5 min at 95˚C | 30 | 60 s at 95˚C | 60 s at 55˚C | 60 s at 72˚C | 10 min at 72˚C | 138 | [26] |
| | st-R | GCAGGCTTACAACACAATTCACAGCAG | | | | | | | | |
| *stx1* | stx1-F | CAACACTGGATGATCTCAG | 5 min at 94˚C | 35 | 60 s at 94˚C | 60 s at 55˚C | 60 s at 72˚C | 10 min at 72˚C | 349 | [22] |
| | stx1-R | CCCCCTCAACTGCTAATA | | | | | | | | |
| *stx2* | stx2-F | ATCAGTCGTCACTCACTGGT | 5 min at 94˚C | 35 | 60 s at 94˚C | 60 s at 55˚C | 60 s at 72˚C | 10 min at 72˚C | 110 | [22] |
| | stx2-R | CTGCTGTCACAGTGACAAA | | | | | | | | |
| *eae* | eae-F | CCCGAATTCGGCACAAGCATAAGC | 5 min at 95˚C | 30 | 60 s at 95˚C | 60 s at 55˚C | 60 s at 72˚C | 10 min at 72˚C | 881 | [27] |
| | eae-R | CCCGGATCCGTCTCGCCAGTATTCG | | | | | | | | |
| *fimA* | fimA-F | CCTTTCTCCATCGTCCTGAA | 2 min at 95˚C | 35 | 30 s at 95˚C | 30 s at 55˚C | 60 s at 72˚C | 5 min at 72˚C | 85 | [28] |
| | fimA-R | TGGTGTTATCTGCCTGACCA | | | | | | | | |
| *stn* | stn-F | CTTTGGTCGTAAAATAAGGCG | 2 min at 95˚C | 35 | 30 s at 95˚C | 30 s at 55˚C | 60 s at 72˚C | 5 min at 72˚C | 260 | [28] |
| | stn-R | TGCCCAAAGCAGAGAGATTC | | | | | | | | |
| *invA* | invA-F | GTGAAATTATCGCCACGTTCGGGCAA | 2 min at 95˚C | 35 | 30 s at 95˚C | 30 s at 58˚C | 60 s at 72˚C | 5 min at 72˚C | 284 | [28] |
| | invA-R | TCATCGCACCGTCAAAGGAACC | | | | | | | | |

## Results

### Phenotype of AMR in *Salmonella* and *E. coli* isolates

The resistance rate of *Salmonella* (*n* = 126) and *E. coli* (*n* = 409) isolates were presented (Table 4). *Salmonella* resistant to at least one antibiotic was reported almost 70% (*n* = 125/129). The prevalence of MDR *Salmonella* was 23.0% (*n* = 29). The most prevalence of AMR in *Salmonella* isolates was sulfamethoxazole (95.2%, *n* = 120/126), followed by trimethoprim (37.3%, *n* = 47/126), and ampicillin (36.5%, *n* = 46/126). The AMR pattern found in *Salmonella* isolates were SUL (37.3%, *n* = 47/126), AMP-SUL-TET-TRI (11.1%, *n* = 14/126), and SUL-TRI (10.3%, *n* = 13/126), respectively.

Of all *E. coli* isolates, 94.1% (*n* = 385/409) were resistant to at least one antibiotic, while the prevalence of MDR was observed at 40.1% (*n* = 164/409). Only 24 *E. coli* isolates (5.9%) were susceptible to all tested antibiotics. The predominant AMR prevalence were sulfamethoxazole (77.8%, *n* = 318/409), ampicillin (55.3%, *n* = 226/409), and tetracycline (40.1%, *n* = 164/409), respectively. The most common resistance patterns of *E. coli* (*n* = 409) were SUL (24.0%, *n* = 98/409), followed by AMP-CHO-STR-SUL-TET-TRI (6.8%, *n* = 28/409), AMP (5.9%, *n* = 24/409), and AMP-SUL (5.4%, *n* = 22/409).

### The presence of AMR genes in *Salmonella* and *E. coli* isolates

The *Salmonella* isolates (*n* = 126) from oyster and estuarine water were harbored *sul3* (14.3%, *n* = 18), followed by $bla_{TEM}$ (11.9%, *n* = 15), *cmlA* (11.9%, *n* = 15), *tetA* (11.1%, *n* = 14), and *dfrA12* (9.5%, *n* = 12), while *catA*, *catB*, *qnrA*, *aac(3)IV*, *aadA1*, *strB*, *tetB*, *sul1*, *sul2*, and *dfrA1* were not observed (Table 5). The $bla_{TEM-1}$ (31.5%, *n* = 129), *tetA* (25.4%, *n* = 104), and *strA* (14.9%, *n* = 61) were predominant resistance genes in the *E. coli* isolates, while *qnrA*, *aac(3)IV*, and *strB* were absent (Table 5).

### ESBL production in *Salmonella* and *E. coli* isolate

For all *Salmonella* isolates, two (1.6%) isolates from oyster samples (serovars Augustenborg and II) (data not shown) were confirmed as ESBL producing isolates. The serovar II of ESBL-producing *Salmonella* was $bla_{TEM-1}$ positive. Eight (2.0%) out of 409 *E. coli* isolates from estuarine water were ESBL-producers, of which three isolates were positive for $bla_{CTX-M-55}$. None of ESBL-producing *E. coli* isolates was retrieved from oyster samples.

**Table 4. Resistance rate of *E. coli* (*n* = 409) and *Salmonella* (*n* = 126) isolates from oyster and estuarine water samples.**

| Antimicrobial agent | Number of AMR isolates (%) | | | | | |
|---|---|---|---|---|---|---|
| | *E. coli* isolates | | | *Salmonella* isolates | | |
| | Oyster (*n* = 250) | Estuarine water (*n* = 159) | Total (*n* = 409) | Oyster (*n* = 123) | Estuarine water (*n* = 3) | Total (*n* = 126) |
| Ampicillin | 124 (49.6%) | 102 (64.2%) | 226 (55.3%) | 46 (37.4%) | - | 46 (36.5%) |
| Chloramphenicol | 38 (15.2%) | 38 (23.9%) | 76 (18.6%) | 13 (10.6%) | - | 13 (10.3%) |
| Ciprofloxacin | 12 (4.8%) | 8 (5.0%) | 20 (4.9%) | 11 (8.9%) | - | 11 (8.7%) |
| Gentamicin | 11 (4.4%) | 9 (5.7%) | 20 (4.9%) | 12 (9.8%) | - | 12 (9.5%) |
| Streptomycin | 75 (30.0%) | 57 (35.8%) | 132 (32.3%) | 11 (8.9%) | - | 11 (8.7%) |
| Sulfamethoxazole | 204 (81.6%) | 114 (71.7%) | 318 (77.8%) | 117 (95.1%) | 3 (100.0%) | 120 (95.2%) |
| Tetracycline | 95 (38.0%) | 69 (43.4%) | 164 (40.1%) | 31 (25.2%) | - | 31 (24.6%) |
| Trimethoprim | 83 (33.2%) | 67 (42.1%) | 150 (36.7%) | 47 (38.2%) | - | 47 (37.3%) |
| Ceftazidime | - | 5 (3.1%) | 5 (1.2%) | 4 (3.3%) | 1 (33.3%) | 5 (4.0%) |
| Cefotaxime | - | 11 (6.9%) | 11 (2.7%) | 23 (18.7%) | 1 (33.3%) | 24 (19.0%) |
| Cefpodoxime | - | 8 (5.0%) | 8 (2.0%) | 3 (2.4%) | - | 3 (2.4%) |

**Table 5. Genotypic characterization of *E. coli* (*n* = 409) and *Salmonella* (*n* = 126) isolated from oysters and estuarine waters.**

| Gene | Number of AMR isolates (%) | | | | | |
|---|---|---|---|---|---|---|
| | *E. coli* isolates | | | *Salmonella* isolates | | |
| | Oyster (*n* = 250) | Estuarine water (*n* = 159) | Total (*n* = 409) | Oyster (*n* = 123) | Estuarine water (*n* = 3) | Total (*n* = 126) |
| *catA* | 4 (1.6%) | 3 (1.9%) | 7 (1.7%) | - | - | - |
| *catB* | 1 (0.4%) | 1 (0.6%) | 2 (0.5%) | - | - | - |
| *cmlA* | 18 (7.2%) | 20 (12.6%) | 38 (9.3%) | 15 (12.2%) | - | 15 (11.9%) |
| *qnrA* | - | - | - | - | - | - |
| *qnrB* | 3 (1.2%) | 2 (1.3%) | 5 (1.2%) | 4 (3.3%) | - | 4 (3.2%) |
| *qnrS* | 25 (10.0%) | 24 (15.1%) | 49 (12.0%) | 10 (8.1%) | - | 10 (7.9%) |
| *aac(3)IV* | - | - | - | - | - | - |
| *aadA1* | 26 (10.4%) | 21 (13.2%) | 47 (11.5%) | - | - | - |
| *strA* | 28 (11.2%) | 33 (20.8%) | 61 (14.9%) | 5 (4.1%) | - | 5 (4.0%) |
| *strB* | - | - | - | - | - | - |
| *tetA* | 54 (21.6%) | 50 (31.4%) | 104 (25.4%) | 14 (11.4%) | - | 14 (11.1%) |
| *tetB* | 10 (4.0%) | 6 (3.8%) | 16 (3.9%) | - | - | - |
| *sul1* | 4 (1.6%) | 1 (0.6%) | 5 (1.2%) | - | - | - |
| *sul2* | 19 (7.6%) | 32 (20.1%) | 51 (12.5%) | - | - | - |
| *sul3* | 19 (7.6%) | 22 (13.8%) | 41 (10.0%) | 18 (14.6%) | - | 18 (14.3%) |
| *dfrA1* | 5 (2.0%) | 3 (1.9%) | 8 (2.0%) | - | - | - |
| *dfrA12* | 17 (6.8%) | 18 (11.3%) | 35 (8.6%) | 12 (9.8%) | - | 12 (9.5%) |
| $bla_{TEM-1}$ | 69 (27.6%) | 60 (37.7%) | 129 (31.5%) | 15 (12.2%) | - | 15 (11.9%) |

## Distribution of virulence genes

Out of 409 *E. coli* isolates, 50.0% of *lt*, 41.1% of *stx1* and 40% of *stx2* were MDR. Of all *Salmonella* isolates, *invA* (77.0%, *n* = 97/126), *stn* (77.0%, *n* = 97/126), and *fimA* (69.0%, *n* = 87/126) were the frequently found virulent genes (Table 6). Among these, the *Salmonella* isolates positive to *invA* (28.9%), *fimA* (29.9%), and *stn* (26.8%) were MDR. The *E. coli* isolates mainly harbored *stx1* (17.8%, *n* = 73/409), followed by *lt* (11.7%, *n* = 48/409) and *stx2* (1.2%, *n* = 5/409) (Table 6). The *E. coli* isolates from oyster samples predominantly contained *stx1* (10.3%) and *lt* (9.5%), while the isolates from estuarine water commonly carried *stx1* (7.6%). The *st* and *stx2* genes were not detected in *E. coli* isolated from estuarine water.

## Co-existence among AMR, ESBL production, and virulence genes

One *Salmonella* isolated from oyster sample harbored $bla_{TEM-1}$ with MDR to ampicillin, chloramphenicol, sulfamethoxazole, trimethoprim, and tetracycline. The latter was also positive to *invA*, *sul3*, *cmlA*, and *dfrA12* genes. An ESBL-producing *E. coli* isolated from cultivation water harbored $bla_{TEM-1}$ was resistant to ampicillin, while three ESBL-producing *E. coli* with $bla_{CTX-M-55}$ were MDR.

## Association between AMR, virulence genes, ESBL production, and MDR

Based on logistic regression analyses, the sulfamethoxazole-resistant *Salmonella* were statistically associated with the presence of ampicillin resistance (OR = 3.06), trimethoprim resistance (OR = 1.47), and *invA* (OR = 1.95) ($p < 0.0001$) compared with those isolates that were not resistant to sulfamethoxazole. The sulfamethoxazole-resistant *Salmonella* were negatively associated with ESBL production ($p < 0.0001$, OR = 0.02) and *stn* ($p < 0.0001$, OR = 0.56).

**Table 6. Virulence genes of *E. coli* (*n* = 409) and *Salmonella* (*n* = 126) isolated from oysters and estuarine waters.**

| Virulent gene | Number of positive isolates (%) | | |
|---|---|---|---|
| | Oyster | Estuarine water | Total |
| ***E. coli*** | | | |
| *lt* | 39 (9.5%) | 9 (2.2%) | 48 (11.7%) |
| *st* | 1 (0.2%) | - | 1 (0.2%) |
| *stx1* | 42 (10.3%) | 31 (7.6%) | 73 (17.8%) |
| *stx2* | 5 (1.2%) | - | 5 (1.2%) |
| *eae* | - | 1 (0.2%) | 1 (0.2%) |
| ***Salmonella*** | | | |
| *invA* | 94 (74.6%) | 3 (2.4%) | 97 (77.0%) |
| *fimA* | 84 (66.7%) | 3 (2.4%) | 87 (69.0%) |
| *stn* | 94 (74.6%) | 3 (2.4%) | 97 (77.0%) |

The *E. coli* isolates resistant to sulfamethoxazole were positively associated with the presence of trimethoprim ($p = 0.027$, OR = 1.55), ESBL production ($p < 0.0001$, OR = 1.83), MDR ($p = 0.008$, OR = 10.33), *addA1* ($p = 0.002$, OR = 3.05), *strA* ($p = 0.044$, OR = 2.66), and *sul3* ($p < 0.0001$, OR = 8.38) than other isolates that were susceptible to sulfamethoxazole (Table 7). However, the *E. coli* isolates resistant to sulfamethoxazole were negatively associated with two virulence genes, including *lt* ($p = 0.025$, OR = 0.43) and *stx* ($p = 0.022$, OR = 0.41), and *dfrA12* ($p < 0.0001$, OR = 0.10).

## Discussion

One of the main findings of this study is more than 90% of *Salmonella* and *E. coli* from fresh oyster (96.7%; *n* = 119/123 of *Salmonella* and 95.2%; *n* = 238/250 of *E. coli* isolates) and estuarine water (100.0%; *n* = 3/3 of *Salmonella* and 92.5%; *n* = 147/159 of *E. coli* isolates) samples

**Table 7. Logistic regression model of the association between *E. coli* resistance to sulfamethoxazole and resistance phenotype, resistance gene, and virulence genes (*n* = 409) classified by type of samples.**

| Predictor | Odds ratio | Std. Err.[a] | 95% C.I.[b] | *p*-value |
|---|---|---|---|---|
| Ampicillin | 0.20 | 0.20 | 0.16–0.25 | < 0.0001 |
| Gentamicin | 0.16 | 0.14 | 0.03–0.94 | 0.043 |
| Trimethoprim | 1.55 | 0.31 | 1.05–2.30 | 0.027 |
| Cefotaxime | 0.33 | 0.18 | 0.11–0.97 | 0.044 |
| ESBL production | 1.83 | 0.16 | 1.55–2.17 | < 0.0001 |
| MDR | 10.33 | 9.08 | 1.84–57.87 | 0.008 |
| *lt* | 0.43 | 0.16 | 0.21–0.90 | 0.025 |
| *stx* | 0.41 | 0.16 | 0.19–0.88 | 0.022 |
| *addA1* | 3.05 | 1.10 | 1.51–6.18 | 0.002 |
| *strA* | 2.66 | 1.29 | 1.03–6.89 | 0.044 |
| *sul3* | 8.38 | 5.30 | 2.43–28.91 | < 0.0001 |
| *dfrA12* | 0.10 | 0.05 | 0.040–0.27 | < 0.0001 |
| Constant | 4.34 | 2.18 | 1.63–11.61 | 0.003 |

AIC[c] = 344.67

[a]Std. Err. is Standard Error

[b]C.I.: Confidence Interval

[c]AIC: Akaike Information Criteria

were resistant to at least one antimicrobial. MDR *Salmonella* (23.0%) and *E. coli* (40.0%) were also isolated, even though the oysters were received from wild caught with no evidence of antimicrobial use. This cultivation site could be contaminated from nearby communities and agriculture according to previous studies [29, 30], so that trackback investigation to identify the source of AMR in estuarine environment is recommended. Estuarine water was considered a potential hotspot to surveillance of AMR distribution in the environment [30]. Humans can be infected with AMR bacteria by eating aquatic animals or direct contact with contaminated environment. Resistant bacteria found in this study are considered as important estuarine environmental pollutants that can adversely affect food security and public health.

In this study, the highest resistance rates acquired by both *Salmonella* (95.2%) and *E. coli* (77.8%) was to sulfamethoxazole. However, a previous study reported the lower prevalence of *Salmonella* resistant to sulfonamides (56.5%) in retail aquaculture products such as shellfish, calm, fish, shrimp, and others in Shanghai [31]. The high prevalence of sulfamethoxazole observed in this study may be widely used in human and animal medicine because this antimicrobial can be used to treat and prevent many bacterial infections at affordable cost [32].Therefore, it is possible that sulfamethoxazole could disseminate and accumulate to the environment. Sulfamethoxazole is effective against both Gram-negative and Gram-positive bacteria, including *E. coli* and *Listeria monocytogenes*. This antimicrobial agent is commonly used to treat urinary tract infection, bronchitis, and prostatitis. In veterinary medicine, sulfonamides have been used in swine and cattle production for treatment of urinary and respiratory tract infection. High concentration of sulfonamides in the environment has been indicated in livestock manure due to the common use of this antimicrobial [33, 34]. Sulfamethoxazole-resistant bacteria was also found in surface water and soil causing environmental pollutants as a result of the widely used in treatment of animals and humans [35]. The impact of sulfonamides contamination in the environment could result in hazardous to human health (e.g., difficult to treat of resistant bacteria, prolong hospital stay etc.), and alter microbial community [33]. However, the consequences of sulfonamide contamination in the ecosystem were still unclear [36]. Therefore, the removal of these resistant bacteria from healthcare facilities, livestock farms, and communities are needed to reduce the contamination to the coastal environment. Previous studies developed the removal of sulfonamides by using anaerobic membrane bioreactor in swine wastewater, and the use of *Pleurotus eryngii* for degradation of sulfonamides [37, 38].

Besides sulfamethoxazole resistance, the high resistance was observed for trimethoprim (37.3%) and ampicillin (36.5%) in *Salmonella*, and for ampicillin (55.3%), and tetracycline (40.1%) in *E. coli*. These findings agreed with previous studies conducted in aquatic animals and estuarine environment [39, 40]. High resistance rates to ampicillin (100%) and erythromycin (83.33%) in *Salmonella* isolates were previously reported in water and sediment [41]. The high resistance to sulfamethoxazole, ampicillin, tetracycline, and trimethoprim observed in this study was commonly reported in humans and animals [42–45]. In Thailand, the molecular epidemiology and association of AMR among of *E. coli* and *S. enterica* have been extensively investigated from pigs, pork, and humans indicating the potential risk of AMR spreading [43, 46]. Even though the precise genetic relationship information is still lacking, the observations of resistance to these antimicrobials in humans, food-producing animals, and environment in the same country confirm that AMR is a complex One Health issue.

*S. enterica* serovar Paratyphi B causes a serious disease, Paratyphoid, in humans. The serovars Paratyphi B poses a significant health risk due to being associated with sporadic outbreaks of human infection and multistage outbreaks of seafood products [46–48]. The symptoms of paratyphoid infection in humans are fever, loss of appetite, weakness, headaches, diarrhea, and may be a life-threatening multi-systemic illness. The pathogens were recently isolated from

poultry and poultry meat from Europe and Latin America [49]. A study reported that serovar Paratyphi B was isolated from oysters (22.7%) in Thailand [15]. In this study, the serovars Paratyphi B was isolated from oysters (13.5%, $n = 17/126$), and all these isolates were resistant to at least one antimicrobial and 29.4% ($n = 5/17$) were MDR. More than 75% ($n = 13/17$) of these isolates contained virulence genes (i.e., *fimA* and *stn*), and 64.7% ($n = 11/17$) of all Paratyphi B isolates harbored *invA*. The presence of MDR Paratyphi B isolates in oysters may pose a serious threat to public health in the near future due to the difficulty in controlling strategic action.

Most resistance genes detected in this study corresponded well to observed resistance phenotype, suggesting that resistance genes were usually expressed when present. In *Salmonella* isolates, the most detected resistance genes were *sul3* (14.3%), *bla*TEM (11.9%), *cmlA* (11.9%), and *tetA* (11.1%), while those in *E. coli* isolates were *bla*TEM (31.5%), followed by *tetA* (25.4%) and *strA* (14.9%). High prevalence (91.3%) of *bla*TEM gene was previously reported in oysters [50], which agreed with this study. This study observed the presence of β-lactamase encoding *bla*TEM-1 indicating a narrow spectrum activity against β-lactamase of *E. coli* and *Salmonella*. This indicated that the estuarine environment serves as a potential hotspot of AMR bacteria carrying resistance determinants that may be transferred to bacterial pathogens in humans and animals.

In this study, the occurrence of ESBL-producing *E. coli* (2.0%) and *Salmonella* (1.6%) was lower than in a previous study, which greatly varied in humans (11–72%), animals (0–72%), and wastewater (7–79%) in West and Central Africa [51]. Greater than 40% of wastewater from Tunisia were positive to ESBL-producing *Enterobacteriaceae* [52]. In this study, *bla*TEM-1 ($n = 2$) and *bla*CTX-M-55 ($n = 3$) were reported with MDR, which agreed with previous studies in aquatic environment and migratory birds [53, 54]. Furthermore, the *bla*TEM and *bla*CTX-M isolates were the common widespread genes from wild fish and aquatic environment [54, 55]. More specifically, the *bla*TEM-1, *bla*CTX-M-14 and *bla*CTX-M-15 genes were reported from marine bivalve mollusks [8]. Even though the low rates of ESBL producing bacteria were observed in this study, the positive ESBL isolates were commonly identified MDR bacteria. Hence, the occurrence of ESBL producing bacteria that harbored MDR signifies the public health threat.

The association between resistance to sulfamethoxazole and other predictors, including AMR, MDR, virulence genes, and ESBL production were examined under the logistic regression models (Tables 7, 8). The complexity of association among resistance and virulence of *E. coli* and *Salmonella* was observed. Sulfamethoxazole resistance in *E. coli* was positively associated with trimethoprim resistance, ESBL production, MDR, and the presence of *addA1*, *strA*,

**Table 8. Logistic regression model of the association between *Salmonella* resistance to sulfamethoxazole and resistance phenotype, resistance gene, and virulence genes ($n = 126$) classified by type of samples.**

| Predictor | Odds ratio | Std. Err.[a] | 95% C.I.[b] | *p*-value |
|---|---|---|---|---|
| Ampicillin | 3.06 | 0.20 | 2.70–3.47 | < 0.0001 |
| Trimethoprim | 1.47 | 0.070 | 1.34–1.62 | < 0.0001 |
| ESBL production | 0.02 | 0.0002 | 0.017–0.018 | < 0.0001 |
| *invA* | 1.95 | 0.14 | 1.69–2.26 | < 0.0001 |
| *stn* | 0.56 | 0.006 | 0.55–0.57 | < 0.0001 |
| Constant | 15.43 | 0.41 | 14.65–16.26 | < 0.0001 |

AIC[c] = 44.72

[a]Std. Err. is Standard Error

[b]C.I.: Confidence Interval

[c]AIC: Akaike Information Criteria

and *sul3*, but these isolates were negatively associated with *lt*, *stx*, and *dfrA12*. The major concern of these findings was almost half of *E. coli* carrying virulence genes were MDR bacteria. A co-selection of resistance and virulence can occur through mobile genetic elements such as integrons, transposons, and integrative conjugative elements [56]. The infection of resistant and virulent pathogens is detrimental to human health since they cause difficulty to treat and increase treatment failure. On the other hand, sulfamethoxazole resistance in *Salmonella* was positively correlated with resistance to ampicillin and trimethoprim, and *invA*, but they were negatively associated with ESBL production and *stn*. This finding indicated the complexity of AMR, virulence factors and resistance determinants in the environment. A quarter of *Salmonella* carrying virulence genes were MDR. Thus, sulfamethoxazole resistance isolates can co-selection to many classes of antimicrobials, virulence genes, and ESBL production. A previous study indicated that resistance and virulence plasmids were linked simultaneously [57]. As a result, the infection of resistant and virulent bacteria may cause more complicated treatment and increase morbidity and mortality rates due to failure of bacterial treatment.

Shiga toxin is bacterial exotoxin related to highly cytotoxic class II ribosome [58]. In this study, *stx1*, Shiga toxin-producing *E. coli* (STEC) was most frequently found in oysters and estuarine waters, while *eae* gene representing enteropathogenic *E. coli* (EPEC) was reported in estuarine water at a low rate (0.2%). A previous study indicated that none of virulence genes related to STEC and EPEC were identified in oysters and mussels from Atlantic Canada [29], in contrast to the results in this study. Wildlife and aquaculture, including fish and shellfish have been identified as one of important sources of STEC spillover from livestock animals [59]. The high rate of *stx1* in this study raise public health concerns of seafood safety, since major clinical signs of STEC infection in humans are bloody diarrhea, hemorrhagic colitis, and hemolytic uremic syndrome, and may be life-threatening.

The *fimA*, *stn*, and *invA* genes are common virulence genes that play an important role in the pathogenicity of *Salmonella* infection. The *fimA* gene is a common structural subunit of type 1 fimbrial protein, while *stn* is heat-labile *Salmonella* enterotoxin affecting epithelial cells [60, 61]. The *invA* gene is an important structural component of *Salmonella* pathogenicity island, which is related to invasion of gut epithelial tissues in human and animals [28]. In this study, 77.0% of *Salmonella* isolates were positive to *invA* gene, even though this gene has been used for confirmation of *Salmonella* in food animals. This agreed with previous studies where the absence of *invA* gene was found in poultry production [62, 63]. In seafood and environmental samples, some *Salmonella* isolates confirmed with biochemical test did not contain *invA* gene [64–66]. The absence of *invA* gene may be because *Salmonella* was not invasive or had other invasive mechanisms [67]. However, the absence of *invA* genes is a rare occasion. The combination of PCR and next generation sequencing (NGS) is proposed to increase sensitivity of *Salmonella* detection of resistance in environmental samples [68].

In conclusion, MDR and ESBL-producing *E. coli* are widespread in the estuarine environment, highlighting the need for continuing AMR monitoring programs in shellfish harvested area. Knowing the magnitude of AMR circulated in the environment can facilitate developing strategic action plans to mitigate the possible transmission of resistance bacteria among humans, animals, and environment. In addition to phenotypic detection of AMR, identification of AMR driving sources and monitoring of genetic information of resistance organisms are required to better understanding reduce the occurrence and transference of AMR in aquatic animals and estuarine waters. Oysters and estuarine water serve as overlooked natural reservoirs of AMR contamination. Awareness of seafood safety and increase personal hygiene are suggested to reduce AMR infection from seafood consumption.

## Acknowledgments

The author would like to thank Saran Anuntawirun for laboratory assistance.

## Author Contributions

**Conceptualization:** Saharuetai Jeamsripong, Rungtip Chuanchuen.

**Data curation:** Saharuetai Jeamsripong, Varangkana Thaotumpitak.

**Formal analysis:** Saharuetai Jeamsripong.

**Funding acquisition:** Saharuetai Jeamsripong.

**Investigation:** Saharuetai Jeamsripong.

**Methodology:** Mullika Kuldee, Varangkana Thaotumpitak.

**Resources:** Saharuetai Jeamsripong, Rungtip Chuanchuen.

**Supervision:** Saharuetai Jeamsripong.

**Validation:** Saharuetai Jeamsripong.

**Visualization:** Saharuetai Jeamsripong.

**Writing – original draft:** Saharuetai Jeamsripong, Mullika Kuldee, Varangkana Thaotumpitak.

**Writing – review & editing:** Saharuetai Jeamsripong, Rungtip Chuanchuen.

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
