## [Decision Letter · Decision Letter 0]

7 Dec 2022

PONE-D-22-28928Antimicrobial Resistance, Extended-Spectrum β-Lactamase Production and Virulence Genes in Salmonella enterica and Escherichia coli Isolates from Estuarine EnvironmentPLOS ONE

Dear Dr. Jeamsripong,

Thank you for submitting your manuscript to PLOS ONE. After careful consideration, we feel that it has merit but does not fully meet PLOS ONE’s publication criteria as it currently stands. Therefore, we invite you to submit a revised version of the manuscript that addresses the points raised during the review process.

Please address all comments of the reviewer point by point. 

We look forward to receiving your revised manuscript.

Kind regards,

Iddya Karunasagar

Academic Editor

PLOS ONE

Journal Requirements:

Additional Editor Comments:

Please see the reviewer comments. The nucleotide sequences should be deposited in a public database and accession number indicated in the manuscript. There are other portions of the manuscript that need improvement accirding to referee comments.

Reviewers' comments:

Reviewer's Responses to Questions

**Comments to the Author**

1. Is the manuscript technically sound, and do the data support the conclusions?

Reviewer #1: No

2. Has the statistical analysis been performed appropriately and rigorously? 

Reviewer #1: I Don't Know

3. Have the authors made all data underlying the findings in their manuscript fully available?

Reviewer #1: No

4. Is the manuscript presented in an intelligible fashion and written in standard English?

Reviewer #1: Yes

5. Review Comments to the Author

Reviewer #1: The study described a statistical model of the association between the most common AMR phenotype/ genotype, virulence genes among Salmonella and E. coli isolated from Bivalve mollusks, acting as an indicator for environmental faecal contamination and estuarine water. The study is interesting but the beta-lactamase sequences obtained should be submitted into the database and the accession numbers should be added in the manuscript without which the manuscript cannot be considered further.

Line 38. Please replace the old reference with current global human mortality rate attributed to AMR

Line 86. ‘not exposed to antimicrobials’

Line 90. Did you conduct any qualitative survey to know the kind of antibiotics used in human and veterinary medicine?

Line 134. ‘invasion’ should be replaced with ‘invasin’

Line 135-138. PCR conditions, primer sequences or reference should be added

Line 183. TEM-1 cannot be considered as ESBL

Line 185. CTX-M types should be mentioned obtained in nucleotide sequencing

Line 231. Did the local people use estuarine water for drinking/domestic purpose?

Line 239. How did the authors know that sulfamethoxazole is commonly used in human and veterinary medicine in local settings?

6. PLOS authors have the option to publish the peer review history of their article (what does this mean?). If published, this will include your full peer review and any attached files.

Reviewer #1: No

---

## [Author Response · Author response to Decision Letter 0]

20 Jan 2023

Antimicrobial Resistance, Extended-Spectrum β-Lactamase Production and Virulence Genes in Salmonella enterica and Escherichia coli Isolates from Estuarine Environment

Comment Response

Reviewer #1: The study described a statistical model of the association between the most common AMR phenotype/ genotype, virulence genes among Salmonella and E. coli isolated from Bivalve mollusks, acting as an indicator for environmental faecal contamination and estuarine water. The study is interesting but the beta-lactamase sequences obtained should be submitted into the database and the accession numbers should be added in the manuscript without which the manuscript cannot be considered further. The beta-lactamase sequences were deposited in NCBI and the accession number OQ282894-OQ282896 was added to the manuscript in lines 130-133 as follows: “The result of the DNA sequence was blasted and aligned with reference embedded in GenBank database available from the National Centre for Biotechnology Information (NCBI) (http://www.ncbi.nlm.nih.gov/ BLAST) (accession number OQ282894-OQ282896)”.

Line 38. Please replace the old reference with current global human mortality rate attributed to AMR The number of infections and deaths due to antimicrobial resistance was updated and the new reference was replaced. 

Reference 

CDC. Antibiotic resistance. Centers for Disease Control and Prevention, National Center for Emerging and Zoonotic Infectious Diseases (NCEZID), Division of Healthcare Quality Promotion (DHQP). https://www.cdc.gov/drugresistance/about.html. 2022.

Line 86. ‘not exposed to antimicrobials’ The error has been corrected as suggested. 

Line 90. Did you conduct any qualitative survey to know the kind of antibiotics used in human and veterinary medicine? We did not conduct a survey on antimicrobial use, but only addressed the list of antimicrobials used in this study. However, the sentence in lines 90-91 was deleted. 

Line 134. ‘invasion’ should be replaced with ‘invasin’ The word “invasion” was replaced as suggested.

Line 135-138. PCR conditions, primer sequences or reference should be added. PCR conditions, primer sequences, and reference were already added in Tables 1-3. 

cannot be considered as ESBL BlaTEM was removed because it was not considered an ESBL positive isolate.

Line 185. CTX-M types should be mentioned obtained in nucleotide sequencing The CTX-M nucleotide sequence was added in lines 187-188 as follow “Eight (2.0%) of 409 E. coli isolates from estuarine water were ESBL producers, of which three isolates were positive for blaCTX-M-55.” 

Line 231. Did the local people use estuarine water for drinking/domestic purpose? In this study site, locals use estuarine water only for aquaculture. Therefore, lines 234-237 were modified as “Humans can be infected with AMR bacteria by eating aquatic animals or direct contact with contaminated environment”. 

Line 239. How did the authors know that sulfamethoxazole is commonly used in human and veterinary medicine in local settings? The sentence “The high prevalence of sulfamethoxazole observed in this study may be widely used in human and animal medicine because this antimicrobial can be used to treat and prevent many bacterial infections at affordable cost [31].” and reference is added for clarification in lines 243-244. 

Reference 

31. Kemnic TR, Coleman M. Trimethoprim Sulfamethoxazole. In: StatPearls. Treasure Island (FL): StatPearls Publishing; 2022 Jan; https://www.ncbi.nlm.nih.gov/books/NBK513232/.

---

## [Decision Letter · Decision Letter 1]

23 Feb 2023

PONE-D-22-28928R1Antimicrobial Resistance, Extended-Spectrum β-Lactamase Production and Virulence Genes in Salmonella enterica and Escherichia coli Isolates from Estuarine EnvironmentPLOS ONE

Dear Dr. Jeamsripong,

Thank you for submitting your manuscript to PLOS ONE. After careful consideration, we feel that it has merit but does not fully meet PLOS ONE’s publication criteria as it currently stands. Therefore, we invite you to submit a revised version of the manuscript that addresses the points raised during the review process.

 The reviewer has noticed that the accession numbers cited in pages 130-133 are not found in GenBank NCBI database. Please give the correct accession numbers that can be accessed by the readers. 

We look forward to receiving your revised manuscript.

Kind regards,

Iddya Karunasagar

Academic Editor

PLOS ONE

Journal Requirements:

Additional Editor Comments (if provided):

Please see reviewer comments. Please give the correct accession number so that readers can access them.

Reviewers' comments:

Reviewer's Responses to Questions

**Comments to the Author**

1. If the authors have adequately addressed your comments raised in a previous round of review and you feel that this manuscript is now acceptable for publication, you may indicate that here to bypass the “Comments to the Author” section, enter your conflict of interest statement in the “Confidential to Editor” section, and submit your "Accept" recommendation.

Reviewer #1: (No Response)

2. Is the manuscript technically sound, and do the data support the conclusions?

Reviewer #1: Yes

3. Has the statistical analysis been performed appropriately and rigorously? 

Reviewer #1: I Don't Know

4. Have the authors made all data underlying the findings in their manuscript fully available?

Reviewer #1: Yes

5. Is the manuscript presented in an intelligible fashion and written in standard English?

Reviewer #1: Yes

6. Review Comments to the Author

Reviewer #1: The authors have addressed most of the comments but the accession numbers (OQ282894-OQ282896) as mentioned in Line 130-133 is not found in the NCBI-GenBank database!

7. PLOS authors have the option to publish the peer review history of their article (what does this mean?). If published, this will include your full peer review and any attached files.

Reviewer #1: **Yes: **Indranil Samanta

---

## [Author Response · Author response to Decision Letter 1]

6 Mar 2023

The reviewer has noticed that the accession numbers cited in pages 130-133 are not found in GenBank NCBI database. Please give the correct accession numbers that can be accessed by the readers. 

Now, the accession number cited on pages 130-133 has been published in GenBank database (http://www.ncbi.nlm.nih.gov/ BLAST) (accession number OQ282894-OQ282896).

---

## [Decision Letter · Decision Letter 2]

7 Mar 2023

Antimicrobial Resistance, Extended-Spectrum β-Lactamase Production and Virulence Genes in Salmonella enterica and Escherichia coli Isolates from Estuarine Environment

PONE-D-22-28928R2

Dear Dr. Jeamsripong,

We’re pleased to inform you that your manuscript has been judged scientifically suitable for publication and will be formally accepted for publication once it meets all outstanding technical requirements.

Kind regards,

Iddya Karunasagar

Academic Editor

PLOS ONE

Additional Editor Comments (optional):

All comments have been addressed.

Reviewers' comments:

Reviewer's Responses to Questions

**Comments to the Author**

1. If the authors have adequately addressed your comments raised in a previous round of review and you feel that this manuscript is now acceptable for publication, you may indicate that here to bypass the “Comments to the Author” section, enter your conflict of interest statement in the “Confidential to Editor” section, and submit your "Accept" recommendation.

Reviewer #1: (No Response)

2. Is the manuscript technically sound, and do the data support the conclusions?

Reviewer #1: (No Response)

3. Has the statistical analysis been performed appropriately and rigorously? 

Reviewer #1: (No Response)

4. Have the authors made all data underlying the findings in their manuscript fully available?

Reviewer #1: (No Response)

5. Is the manuscript presented in an intelligible fashion and written in standard English?

Reviewer #1: (No Response)

6. Review Comments to the Author

Reviewer #1: (No Response)

7. PLOS authors have the option to publish the peer review history of their article (what does this mean?). If published, this will include your full peer review and any attached files.

Reviewer #1: No

---

## [Editor Report · Acceptance letter]

20 Apr 2023

PONE-D-22-28928R2 

Antimicrobial Resistance, Extended-Spectrum β-Lactamase Production and Virulence Genes in *Salmonella enterica* and *Escherichia coli* Isolates from Estuarine Environment 

Dear Dr. Jeamsripong:

I'm pleased to inform you that your manuscript has been deemed suitable for publication in PLOS ONE. Congratulations! Your manuscript is now with our production department. 

Kind regards, 

on behalf of

Dr. Iddya Karunasagar 

Academic Editor

PLOS ONE